# Validating a cassava production spatial disaggregation model in sub-Saharan Africa

**Kirsty L. Hassall**[1], **Vasthi Alonso Chávez**[2‡]*, **Hadewij Sint**[3], **Joseph Christopher Helps**[2],
**Phillip Abidrabo**[4], **Geoffrey Okao-Okuja**[4], **Roland G. Eboulem**[5], **William J-L. Amoakon**[5],
**Daniel H. Otron**[5], **Anna M. Szyniszewska**[6‡]

1 Inteligent Data Ecosystems, Rothamsted Research, Harpenden, Hertfordshire, United Kingdom, 2 Net
Zero and Resilient Farming, Rothamsted Research, Harpenden, Hertfordshire, United Kingdom, 3 Net Zero
and Resilient Farming, Rothamsted Research, North Wyke, Okehampton, Devon, United Kingdom,
4 National Crops Resources Research Institute, Kampala, Uganda, 5 The Central and West African Virus
Epidemiology (WAVE), Université Félix Houphouët-Boigny, Abidjan, Côte d'Ivoire, 6 CABI, Nosworthy Way,
Wallingford, United Kingdom

‡ VAC and AMS are joint senior authors on this work.
* vasthi.alonso-chavez@rothamsted.ac.uk

journal.pone.0312734

Indies at Cave Hill, BARBADOS

**Data Availability Statement:** All data and code
associated with this paper are openly available. The
original survey data stored as cassava field
perimeters and associated characteristics is

## Abstract

Cassava is a staple in the diet of millions of people in sub-Saharan Africa, as it can grow in
poor soils with limited inputs and can withstand a wide range of environmental conditions,
including drought. Previous studies have shown that the distribution of rural populations is
an important predictor of cassava density in sub-Saharan Africa's landscape. Our aim is to
explore relationships between the distribution of cassava from the cassava production dis-
aggregation models (CassavaMap and MapSPAM) and rural population density, looking at
potential differences between countries and regions. We analysed various properties of cas-
sava cultivations collected from surveys at 69 locations in Côte d'Ivoire and 87 locations in
Uganda conducted between February and March 2018. The relationships between the pro-
portion of surveyed land under cassava cultivation and rural population and settlement data
were examined using a set of generalized additive models within each country. Information
on rural settlements was aggregated around the survey locations at 2, 5 and 10 km circular
buffers. The analysis of the original survey data showed no significant correlation between
rural population and cassava production in both MapSPAM and CassavaMap. However, as
we aggregate settlement buffers around the survey locations using CassavaMap, we find
that at a large scale this model does capture large-scale variations in cassava production.
Moreover, through our analyses, we discovered country-specific spatial trends linked to
areas of higher cassava production. These analyses are useful for validating disaggregation
models of cassava production. As the certainty that existing cassava production maps
increases, analyses that rely on the disaggregation maps, such as models of disease
spread, nutrient availability from cassava with respect to population in a region, etc. can be
performed with increased confidence. These benefit social and natural scientists, policy-
makers and the population in general by ensuring that cassava production estimates are
increasingly reliable.

available at https://doi.org/10.6084/m9.figshare.
23657391.v1 Processed survey data in tabular
format is available at https://doi.org/10.6084/m9.
figshare.26983603.v1 Associated code for all
analyses presented in this study is available on
Zenodo under initial release https://doi.org/10.
5281/zenodo.13748021.

**Funding:** This research was supported by the
project "Epidemiological modelling of simultaneous
control of multiple cassava virus diseases" funded
by the Biotechnology and Biological Sciences
Research Council (GCRF-BBSRC) grant number
BB/P022480/1 that funded VAC, KLH and HS
contributions and the project "Validating cassava
distribution map for sub-Saharan Africa to enhance
its impact on effectiveness of surveillance, cassava
disease management and control strategies"
Global Challenges Research Fund – Impact
Acceleration Award (GCRF-IAA), BBSRC project
(S6166) that funded AMS, JH, PA, GO-O, RGE,
WJ-LA and DHO contributions. VAC, KLH and JH
from Rothamsted Research acknowledge support
from the Growing Health Institute Strategic
Programme (BBS/E/RH/230003C). The funders
had no role in study design, data collection and
analysis, decision to publish, or preparation of the
manuscript.

**Competing interests:** The authors have declared
that no competing interests exist.

# 1. Introduction

*Manihot esculenta* (Euphorbiaceae), commonly known as cassava, is a perennial vegetatively propagated tuber crop with a high calorific content. Cassava is endemic to Brazil but has become a staple in Africa following its introduction to the continent in the 16[th] century, where it is now grown both for subsistence and as a cash crop for direct sale and industrial applications [1]. Beyond South America and Africa, it is also widely cultivated in southeast Asia, where Thailand is the biggest producer followed by Indonesia [2]. Today, cassava is grown in more than 39 African and 56 other countries around the world [1] and has become the staple food crop of approximately 800 million people worldwide [3]. The total worldwide production of cassava was about 303 million metric tons in 2019 with Nigeria being the world's largest cassava producer and Africa contributing to approximately 63% of the global production [2]. The widespread cultivation of cassava can be attributed to the flexibility of planting season and harvest, its high drought tolerance, and its ability to grow even in poor soil conditions [3]. Additionally, while many other crops are projected to be negatively impacted by climate change in Africa, cassava is one of the few crops that is expected to benefit from it [4].

Despite the importance of cassava as a staple crop, there is a lack of verified information describing the spatial distribution and density of cassava cultivation. Improved representation of cassava cultivation spatially would enable more targeted surveillance and management planning for devastating cassava pests and pathogens, including cassava mosaic disease (CMD), cassava brown streak disease (CBSD), cassava bacterial blight (CBB), cassava mealybug and fungal pathogens causing root rot [5–9]. Each of these diseases can cause significant yield losses, with CMD and CBSD able to lead to between 30–40% yield losses in Africa, and up to 70% yield loss [10]. It would also enhance the monitoring and prediction of pathogen spread and the planning of pest and disease control strategies such as the dissemination of clean seeds and deployment of improved varieties.

One challenge in accurately mapping the cultivation of cassava is results from the highly flexible planting and harvesting patterns of smallholder cassava growers. Small field sizes and frequent intercropping pose continued challenges in mapping cassava using satellite imagery. As cassava is both a subsistence and cash crop requiring relatively low inputs, it is often grown in rural areas. Previous studies (Carter & Jones, 1993; Herrera Campo et al., 2011; Szyniszewska, 2020; Ugwu & Nweke, 1996) have shown that socioeconomic and demographic properties, including the density of rural population, are important predictors of cassava density in sub-Saharan Africa's landscape [11–14].

Consequently, one method that has been used to produce more precise information on the cassava spatial distribution is the use of disaggregation models, which take coarse indicators, such as yield information for individual provinces and rural population density maps, to predict the spatial distribution of crops at finer scales. Two such models, which we study in this paper, are the Spatial Production Allocation Model MapSPAM [15–17] and CassavaMap [14]. MapSPAM was first developed to derive estimates of 8 crops in Brazil at a resolution of 25–100 square kilometers [18], but has since been expended to include 42 crop types at a 5 arcmin resolution [19]. The MapSPAM cassava distribution layer represents a disaggregation of the crop production statistics using various inputs, including irrigation masks, cropland and rural population distributions, and crop biophysical suitability indices. The disaggregation outputs from MapSPAM were produced simultaneously for 42 crops including cassava, using an entropy-based data-fusion approach [15–17]. CassavaMap specifically illustrates cassava production density for the year 2014 on an approximately 1 km x 1 km spatial resolution [14]. This model disaggregates sub-national crop production statistics, operating on the primary

assumption that the rural population is the strongest predictor of cassava cultivation distribution in Africa [14] as defined by the LandScan 2014 [20] population density layer [15].

In this study, we developed and carried out surveys in cassava-growing regions of Côte d'Ivoire and Uganda to 1) quantify the characteristics of cassava cultivation across distinct cassava-growing regions, 2) to corroborate or discard the hypothesis that directly links rural population and cassava density, 3) to find out how the cassava density in the surveys correlates with two existing cassava cultivation density models, and 4) to investigate the driving influences in the observed mismatch between surveyed data and point predictions from CassavaMap. For the survey data collection, we used the ArcGIS Collector app to aid the measurement of the extent of the survey locations grids [21] and for the data and statistical analyses, we used the R programming language [22] due to their ability to produce the desired analyses, ease of use and accessibility.

In both countries, the northern parts experience a hotter, semi-arid climate. In contrast, the southern regions have more humid, tropical climate, supporting dense vegetation and agriculture. As both countries represent a variety of agro-climatological zones they provide insight into the patterns of cassava cultivation in various climates.

## 2. Materials and methods

### 2.1 Data sources

**2.1.1 Cassava density survey.** The cassava cultivation surveys obtained information from 69 locations in Côte d'Ivoire and 96 locations in Uganda during a total of four weeks of fieldwork conducted in February and March 2018 (Fig 1). A predefined 100 x 100 m² fishnet grid

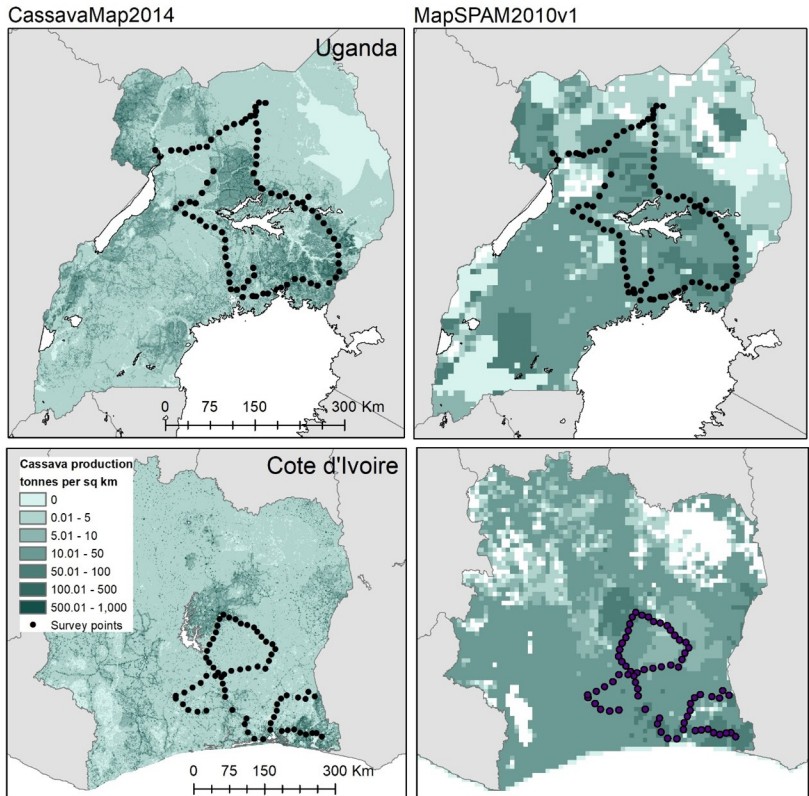

**Fig 1. Illustration of the visited locations in Uganda and Côte d'Ivoire for the cassava density survey over the CassavaMap (left) and the SPAM2010v1 model (right).** Sources: [14, 17].

was set up in the ArcGIS Collector app to aid the measurement of the extent of the survey locations grids [21]. Survey locations were chosen at random at approximately 15–20 km intervals along major motorable roads in each country (Fig 1).

Before accessing the sites, we sought permission from the farmers or village leaders to conduct the survey. The survey locations represented various levels of population density, including rural, suburban, and urban areas.

At each sampling location, the team surveyed an area of approximately 200 x 200 m$^2$ area, consisting of four 100 m x 100 m predefined quadrants. The surveyors recorded the perimeter of all cassava fields within the selected study area, the size of small cassava patches and the number of individual plants grown outside any main field patch. In the following, we use field to mean an area of cassava cultivation with reasonably uniform density within the study area. The team recorded attributes of the individual fields and patches, such as whether the cassava was intercropped, the cassava plants' age, and the density of each field (high, medium, and low density). The density of cassava cultivation was not defined on strict measurements, and rather the subjective experience of surveyors in assessing the planting practices. For intercropped fields, the other crops present in the fields were listed. The locations of inhabited buildings were recorded as point locations within each surveyed quadrant and the approximate building size was recorded. The surveyors could turn on the tracking function which automatically marked the route of the survey team on the ArcGIS Collector screen to ensure the whole area was visited. In areas with access difficulties or safety concerns, for example, in certain suburban areas, only one or two 100 x 100 m quadrants were selected for surveys for practical reasons.

The data collected in the survey were exported and saved as a collection of polygon and point locations [23]. The data were post-processed to calculate the proportion of the study area with cassava fields [24]. The area of the cassava fields was calculated from the perimeter of the fields and patches, and for individual plants, a 0.5 m radius was assumed around each plant.

The total area in cassava production at each survey location $A_C$ was calculated as

$$A_C = \frac{\sum_{i=1}^{M} \alpha_i + \sum_{j=1}^{N} \beta_j + \sum_{k=1}^{K} \gamma_k}{\delta} \tag{1}$$

where $\alpha_i$ is the area of a cassava monoculture field and $M$ is the total number of monoculture fields at the survey location; $\beta_j$ is the area of a cassava intercropped field and $N$ is the total number of intercropped fields at the survey location; $\gamma_k$ is the area of an individual cassava plant and $K$ is the total number of individual plants at the survey location. $\delta$ is the total area of the survey location. A secondary measure of total cassava production was calculated to incorporate i) a lower density of cassava production in intercropped fields (calculated as a weight of 0.75) and ii) the qualitative assessment of cassava density within each field or patch. Specifically, weights $\omega_{i,j}$ were assigned according to Table 1. All other fields with no specific density recording were given a weight of 1.

Thus, the weighted area of cassava production $A_{CW}$ was defined by,

$$A_{CW} = \frac{\sum_{i=1}^{M} \omega_i \alpha_i + 0.75 \cdot \sum_{j=1}^{N} \omega_j \beta_j + \sum_{k=1}^{K} \gamma_k}{\delta} \tag{2}$$

**2.1.2 Cassava production models.** For both CassavaMap and MapSPAM, we extracted predicted cassava density, and additionally from CassavaMap, we extracted harvest area at the point locations of each survey location. We used the 2010 SPAM v1 cassava production and

**Table 1. Assignment of quantitative weights to the qualitative assessment of cassava density within fields and patches as defined by the surveyors.**

| Density | Weight |
|---|:---:|
| Very High | 1.75 |
| High | 1.5 |
| Regular | 0.75 |
| Sparse | 0.5 |
| Very sparse | 0.25 |

harvested area outputs, which are provided at approximately 10?km by 10?km spatial resolution. We compared observed and predicted cassava production by calculating the Spearman rank correlation coefficients using the R package ggcorrplot [25] and by analysing the change in predicted cassava production at survey locations where cassava production was absent and at survey locations where cassava production was present. To investigate the potential for spatial mismatch, we additionally extracted CassavaMap predictions summarised in a buffered region about each survey location.

**2.1.3 Rural population data.** Population distribution data were obtained from LandScan 2014 [20] and a binary mask representing rural settlements from the WorldPop 2018 [26] models. The LandScan 2014 dataset, with a resolution of approximately 1 km by 1 km (~30″ by 30″), was developed as part of the Oak Ridge National Laboratory (ORNL) Global Population Project utilising sub-national census data combined with additional variables such as land cover, roads, urban and rural locations. The census population count data are redistributed according to a weighting scheme [26]. Rural population data (both population density and rural settlements) were extracted at the survey point locations. In addition, these data layers were summarised over buffered regions around each survey location and can be found at [24].

## 2.2 Data processing methods

**2.2.1 Aggregation of buffered data layers.** Aggregation of the information related to variables in the vicinity of the cassava density survey was done using the raster package in R statistical programming software [27]. The buffer data was obtained from the raster layers of the Landscan population data [20], WorldPop settlement data and CassavaMap disaggregation model by extracting values of the raster within specified buffered areas around the sample locations. Specifically, buffer polygons of 2, 5 and 10 km were created around the sample location coordinates. We applied two ways of calculating summary statistics for the buffers around each survey point location. The first approach is to dissolve the buffers, using the function *mask* in R from the raster library, into one object, removing all intersecting areas of the buffers. This was used in the analysis of spatial trends (see Section 2.2.3). The second approach is to keep an individual buffer object (polygon) for each sample point from which general zonal statistics are calculated on the buffered areas and used in the regression modelling (see Section 2.2.2). The summaries of the CassavaMap predictions that were considered were the mean, median, standard deviation, minimum, maximum and lower and upper quartiles. Similarly, summary statistics were calculated at each location for the population data layer and for the settlement data layer, this was restricted to the mean as the settlement information is a binary layer of presence/absence of settlement in each pixel. Aggregated were stored in tabular format and can be found at [24].

**Table 2. Baseline regression models for each variable of interest.** Transformation of the explanatory variable was chosen to best explain the observed relationship. c is a small constant offset calculated as half the minimum non-zero value of the explanatory variable.

| Country | Survey Response Variable (y) | CassavaMap Explanatory Variable (x) | Model |
|---|---|---|---|
| Côte d'Ivoire | Total Cassava Area | Production | $y \sim \log(x + c)$ |
| Côte d'Ivoire | Total Cassava Area | Harvest Area | $y \sim \log(x + c)$ |
| Uganda | Total Cassava Area | Production | $y \sim \log(x + c)$ |
| Uganda | Total Cassava Area | Harvest Area | $y \sim x$ |

**2.2.2 Linking survey data to modelled cassava.** Baseline regression models (Table 2) were used to assess the association between observed cassava production and cassava production predicted from CassavaMap.

No transformation of the response variables was deemed necessary through inspection of the residual plots. Transformation of the explanatory variable was chosen to best explain the observed relationships.

To investigate the impact of the spatial resolution of cassava production and harvested area of CassavaMap predictions along with any potential biases associated with settlement and population density in the surveyed locations, a systematic regression framework (Fig 2) was used for six response variables: total cassava density, total cassava density under monoculture, total cassava density under intercropping and their associated weighted versions. Firstly, to understand the spatial representativeness of CassavaMap, rather than considering the point predictions as an explanatory variable, the extracted aggregated summaries for predicted cassava production density, as listed in S1 Table of S1 Appendix, were each considered in turn. The form of the regression model was constrained to one of four types, 1) a linear relationship, 2) a logarithmic relationship, 3) a quadratic relationship and 4) a non-parametric spline. Secondly, a measure of population density was included (in addition to the measure of predicted cassava) through one of the extracted aggregated summaries as listed in S1 Table of S1 Appendix. The population density variable was constrained to one of four relationships in the model, 1) linear, 2) logarithmic 3) independent non-parametric spline or 4) dependent 2-d non-parametric spline with predicted cassava. Thirdly, a measure of settlement density was included (in

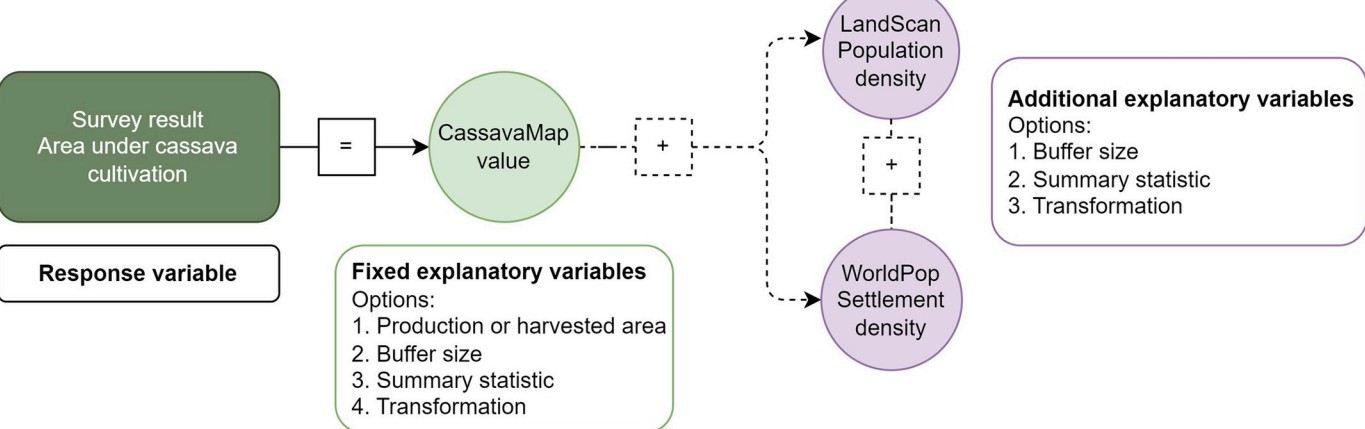

**Fig 2. Illustration of the regression framework to explore the relationships between observed survey data, the predicted cassava density from CassavaMap and settlement and/or rural population density.**

addition to the measure of predicted cassava) through one of the extracted aggregated summaries as listed in S1 Table of S1 Appendix. The settlement density variable was constrained to one of four relationships in the model, 1) linear, 2) logarithmic 3) independent non-parametric spline or 4) dependent 2-d non-parametric spline with predicted cassava. Finally, we considered including measures of both population and settlement density in the model through the relationships described above and an additional 2-d non-parametric spline over both variables.

In total, we explored 31,164 combinations of distinct regression models for each response variable in each country. For each regression model, the Akaike Information Criterion (AIC), Bayesian Information Criterion (BIC) and adjusted $R^2$ were extracted as a measure of model performance.

$$AIC = -logLik + 2p$$

$$BIC = -logLik + \log(n)p$$

$$R^2_{adj} = 1 - \frac{\sum_i (y_i - \hat{y}_i)^2 \Big/ (n-p)}{\sum_i (y_i - \bar{y})^2 \Big/ (n-1)}$$

Where, *logLik* is the log likelihood of the model, *p* is the number of model parameters, *n* is the number of data points included in the regression model, *y* is the data, $\hat{y}$ is the fitted value from the regression model and $\bar{y}$ is the mean of all $y_i$.

The strategy outlined above was used to i) find the best model that explains variation in the survey data of cassava production and ii) to assess the impact of both the distance and type of aggregated summary on predicting cassava production. For the latter, we used an unbalanced ANOVA screening procedure on the extracted AIC from all fitted models. Each of the 31,164 statistical models was associated with particular factors defining what explanatory terms were included, the summary statistic used, and at what buffer distance along with the form of the model (linear vs generalized additive). ANOVA was then used to assess the impact of these distinct factors on the AIC. The ANOVA treatment model is specified by Eq 3.

$$
\begin{aligned}
modeltype * (&cass\_type * population\_type * settlement\_type \\
&+cass\_type * (cass\_dist * cass\_summary) \\
&+population\_type/(population\_dist * population\_summary) \\
&+settlement\_type/settlement\_dist)
\end{aligned}
\tag{3}
$$

where *modeltype* is a binary variable indicating whether the fitted model is a linear model or generalized additive model, *cass_type* is a binary variable indicating whether the cassava model prediction is production or harvest area, *population_type* is a binary variable indicating whether a population covariate has been included or not and *settlement_type* is a binary variable indicating whether a settlement covariate has been included. The terms *_summary* indicate the type of summary statistic used and *_dist* indicates the distance of a buffered region. Point estimates of predictions and population data have a summary = mean and a distance = 0 km.

Due to partial confounding between terms, type II F-statistics were extracted.

Regression models were fitted in R programming language, with generalized additive models fitted using thin-plate regression splines from the *mgcv* package [28, 29] and type II statistics obtained from the *car* package [30].

**2.2.3 Spatial trends.** Geographical trends in the survey data were summarised by i) linear models accounting for administrative regions and ii) generalized additive models along the different transects of the sampling. For both countries, the total cassava area was first log-transformed and separate additive terms were fitted over longitude and latitude independently. There was insufficient data to fit an interaction between the two.

Geographical trends in the CassavaMap predictions are summarised through generalized additive models (GAMs) using the dissolved buffer extraction of the spatial maps of the survey locations to investigate large-scale regional changes. Models were fitted to the natural logarithm of the prediction production with additive smooth terms for longitude and latitude.

## 3. Results

### 3.1 Characteristics of cassava cultivation

**3.1.1 Côte d'Ivoire.** In Côte d'Ivoire, of the 69 visited locations, 52 were found to have some cassava production with 17 having no cassava plants at the time of the survey. Of these 52 locations, 38 included one or more intercropped fields, 46 included monoculture fields and 8 included individual cassava plants outside of a main field area. The number of individual plants at these 8 locations was generally relatively small ranging from 1 to 9. The median (lower and upper quartile) number of cassava fields at each location was 7 (2, 16). The field size was highly skewed with a mean (median) of 557.3 m$^2$ (72.9 m$^2$) for cassava monoculture and 689.2 m$^2$ (142.1 m$^2$) for intercropped cassava fields. The total area allocated to monoculture or intercropped fields was relatively consistent over the 52 locations. In total 221 intercropped fields and 288 monoculture fields were recorded over all surveyed locations. The ratio of intercropped vs. monoculture fields in each location did vary, but in general, categorised into locations either in complete monoculture (27% of locations), complete intercropping (15% of locations) or a 50:50 split (23% of locations), the rest of the locations were represented by an equal mix of intercropping and monoculture fields. A summary of cassava production for the surveyed region is shown in Table 3. The total area under cultivation of cassava per location was highly skewed (Fig 3a and 3b), but similar between cassava in monoculture 3086 m$^2$ (1057 m$^2$) and in intercropped fields 2929 m$^2$ (1511 m$^2$), mean (median) (see Table 3). The type of cassava cultivation (monoculture, intercropping or as individual plants) was generally uncorrelated (Fig 3c).

**3.1.2 Uganda.** In Uganda, of the 87 visited locations, 76 were found to have some cassava production with 11 having no cassava plants at the time of the survey. Of these 76 locations, 57 included intercropped fields, 69 included monoculture fields and 20 included the presence of individual cassava plants. The number of individual plants at these 20 locations was generally quite small ranging from 1 to 6, although 2 locations had 16 or more plants. The median (lower and upper quartile) number of cassava fields at each location was 6 (3, 14). The field size was highly skewed with a mean (median) of 685.8 m$^2$ (322.4 m$^2$) for cassava monoculture and 499.9 m$^2$ (230.9 m$^2$) for intercropped cassava fields. The total area allocated to monoculture or intercropped fields was relatively consistent over the 76 surveyed locations. This corresponds to 297 intercropped fields and 339 monoculture fields recorded over all surveyed locations. The ratio of intercropped vs. monoculture fields in each location did vary, but around 26% of locations demonstrated a general preference for complete monoculture. The total area in cassava production per location was highly skewed (Fig 3d and 3e) with more in monoculture 3059 m$^2$ (1806 m$^2$) than in intercropped fields 1954 m$^2$ (887 m$^2$), mean

**Table 3. Summary of cassava production across surveyed sites in Côte d'Ivoire.**

|  | Cassava present | Intercropping | Monocropping | Individual plants | 50:50 split |
|---|---|---|---|---|---|
| **Locations** | | | | | |
| Locations (n = 69 visited) | 52 | 38 | 46 | 8 | |
| Locations with complete cultivation of this type (%) | | 27% | 15% | | 23% |
| Area cultivated per location—mean (lower, median, upper) | | 2929 m$^2$ (0, 1511, 3322 m$^2$) | 3086 m$^2$ (208, 1057, 4587 m$^2$) | | |
| **Fields** | | | | | |
| Total number of fields across all locations | | 221 | 288 | | |
| No. of cassava fields per location—median (lower and upper quartile) | 7 (2,16) | | | | |
| Field size—mean (lower quartile, median, upper quartile) | | 689.2 m$^2$ (18.7, 142.1, 706.7 m$^2$) | 557.3 m$^2$ (12.9, 72.9, 375.0 m$^2$) | | |

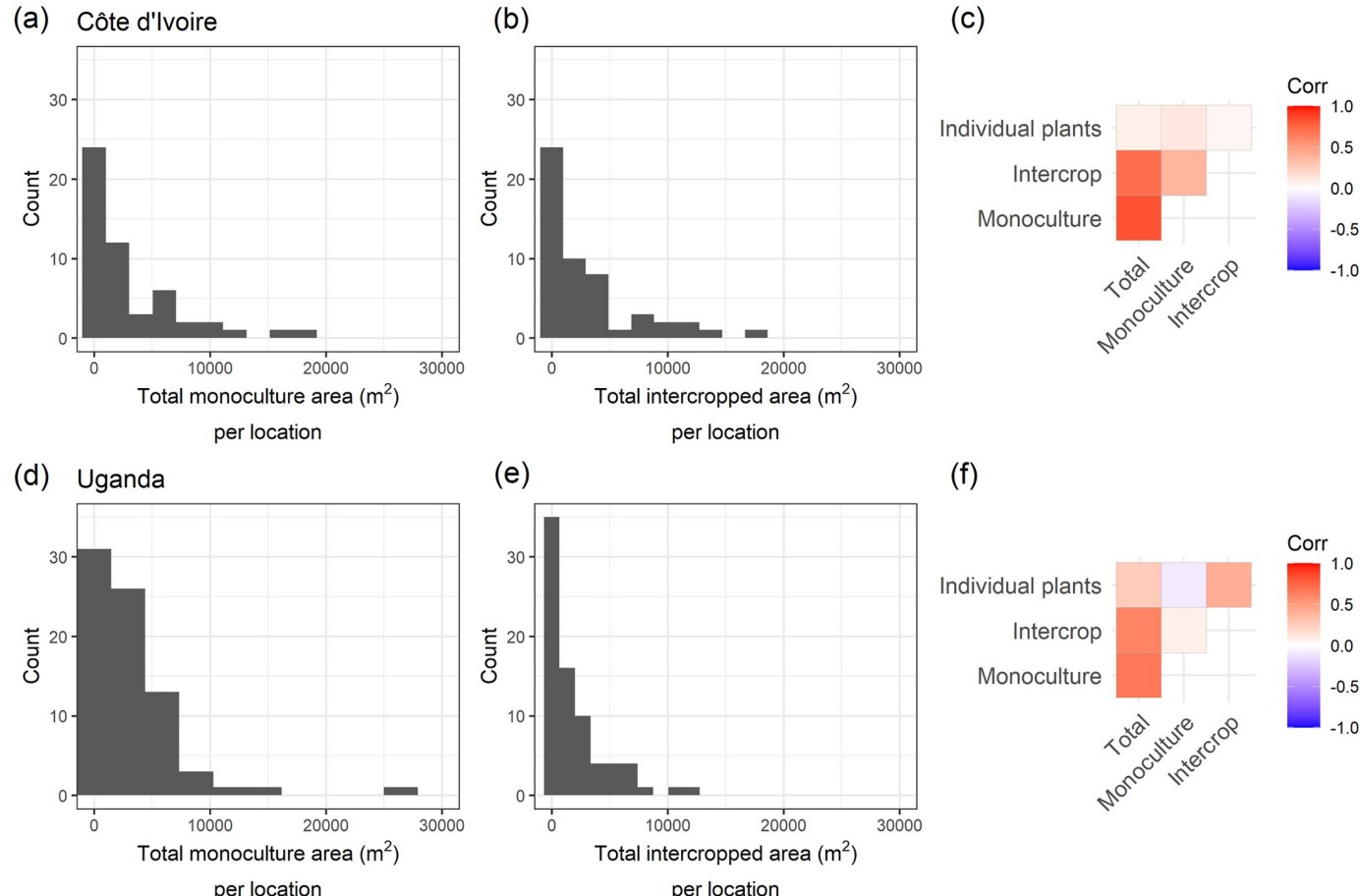

**Fig 3. Histograms of the total area in cassava production at each survey location separated by management system (monoculture vs. intercrop).** The heatmap illustrates the correlation between the total area under different cassava cultivation types (monoculture, intercropping and individual plants). The top row represents results from Côte d'Ivoire and the bottom row from Uganda. Relationships between surveyed cassava density and existing cassava cultivation density model.

**Table 4. Summary of cassava production across surveyed sites in Uganda.**

|  | Cassava present | Intercropping | Monocropping | Individual plants |
|---|---|---|---|---|
| **Locations** | | | | |
| Locations (n = 87 visited) | 76 | 57 | 69 | 20 |
| Locations with complete cultivation of this type (%) | | 9% | 26% | |
| Area cultivated per location—mean (lower quartile, median, upper quartile) | | 1954 m$^2$ (42, 887, 2601 m$^2$) | 3059 m$^2$ (374, 1806, 4356 m$^2$) | |
| **Fields** | | | | |
| Total number of fields across all locations | | 297 | 339 | |
| No. of cassava fields per location—median (lower and upper quartile) | 6 (3,14) | | | |
| Field size mean (lower quartile, median, upper quartile) | | 499.9 m$^2$ (46.2, 230.9, 548.9 m$^2$) | 685.8 m$^2$ (322.4 69.4, 887.5 m$^2$) | |

(median). Intercropped and monoculture cultivation were generally independent, but a positive correlation was observed between the cultivation of individual plants and intercropped fields (Fig 3f). A summary of cassava production for the surveyed region is shown in Table 4.

## 3.2 Linking survey data to rural population density

No detectable relationships were observed between predicted cassava production at the surveyed point locations and the observed cassava area in either Côte d'Ivoire or Uganda (Table 5).

However, investigating if the presence and absence of cassava production in the surveyed locations are related to model predictions (Fig 4), there is some indication of a positive relationship between the presence or absence of cassava production in the surveyed locations and the cassava distribution models predictions.

## 3.3 Spatial trends in surveyed cassava density and CassavaMap predictions

Despite the lack of association between the survey locations and point estimates of the model predictions, larger-scale predictions were investigated through spatial trends. Since the survey was not designed to explore spatial patterns and was restricted to main motorable roads, traditional geostatistics cannot be applied. Instead, we have investigated large-scale directional trends.

**3.3.1 Côte d'Ivoire.** In Côte d'Ivoire, marginal differences in the total area under cassava production across administrative areas were observed ($F_{8,60} = 1.88$, p = 0.08, data square root transformed, Fig 5a). Further, survey areas in the southeast corner and the westerly edge appear to be associated with higher cassava production. This is evident in the predicted smoothed function over longitude (Fig 5b). By extracting the predicted cassava production from CassavaMap at a 10 km buffer around each survey location, similar spatial trends in the longitude could be identified (Fig 5d and 5e), indicating that at larger scales, the CassavaMap predictions capture large-scale variation in cassava production.

**3.3.2 Uganda.** In Uganda, there appear to be "hotspots" of cultivation types across the area, with a higher density of monoculture in the East and a higher density of intercropping in the South region (S1 Fig in S1 Appendix). These differences, however, were not found to be significant in association with the regional boundaries except for the total area under intercropping having a marginal effect ($F_{3,83} = 2.62$, p = 0.056, after square root transformation). In addition, part of the southern survey locations appears to be associated with higher cassava

**Table 5. Spearman rank correlation values between surveyed cassava density and predicted cassava production from CassavaMap and SPAM2010v1.** The upper triangle shows values for Côte d'Ivoire and the lower triangle for Uganda.

| | tot_cassava_area | tot_cassava_area_w | tot_monoculture_area | tot_intercrop_area | CassavaMap_Prod | CassavaMap_HA | MapSPAM2010v1_Prod | MapSPAM2010v1_HA |
|---|---|---|---|---|---|---|---|---|
| tot_cassava_area | | 0.99 | 0.83 | 0.72 | 0.17 | 0.15 | 0.25 | 0.22 |
| tot_cassava_area_w | 0.98 | | 0.86 | 0.69 | 0.19 | 0.16 | 0.26 | 0.23 |
| tot_monoculture_area | 0.68 | 0.78 | | 0.38 | 0.26 | 0.24 | 0.26 | 0.24 |
| tot_intercrop_area | 0.62 | 0.51 | 0.08 | | 0.10 | 0.09 | 0.02 | -0.02 |
| CassavaMap_Prod | 0.13 | 0.13 | 0.10 | 0.12 | | 0.99 | 0.51 | 0.46 |
| CassavaMap_HA | 0.08 | 0.10 | 0.08 | 0.09 | 0.93 | | 0.53 | 0.48 |
| MapSPAM2010v1_Prod | 0.38 | 0.39 | 0.27 | 0.26 | 0.56 | 0.38 | | 0.97 |
| MapSPAM2010v1_HA | 0.37 | 0.39 | 0.24 | 0.26 | 0.55 | 0.45 | 0.84 | |

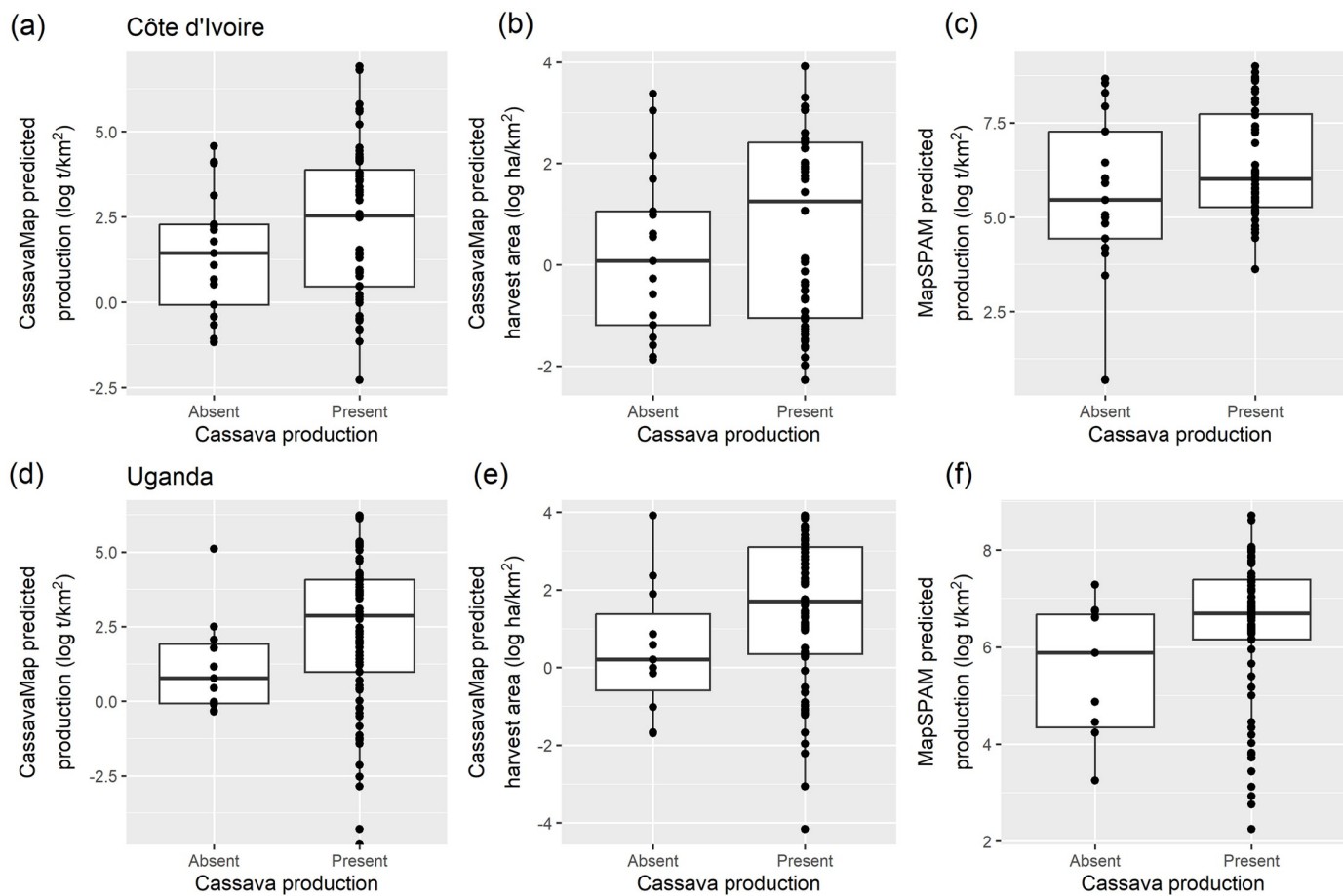

**Fig 4.** Boxplots of model predictions a) CassavaMap production, b) CassavaMap harvest area and c) SPAM2010v1 compared to the presence or absence of cassava production in the surveyed locations. The top row is Côte d'Ivoire, and the bottom row is Uganda.

production. This is evident in the predicted smoothed function over latitude seen in Fig 6f. Predicted cassava production in 10 km radius buffers around each survey location, yielded similar spatial trends in the latitude, indicating that at larger scales, the CassavaMap captures large-scale variation in cassava production in Uganda like in the case of Côte d'Ivoire.

S2 and S3 Tables in S1 Appendix show the ANOVA results from analysing cassava production variables across distinct administrative regions in Côte d'Ivoire and Uganda, respectively.

### 3.4 Impact of settlement and population information on the association between cassava density survey and CassavaMap predictions

Through the extensive regression framework outlined in Section 2.2.2, we investigated the impact of including each data layer and the form with which this should be included, the type of summary statistic used and the buffer distance for each of the different response variables collected from the survey data. Results tables are shown in S4 and S5 Tables of S1 Appendix.

The type of regression model used (linear versus generalized additive) is hugely influential in lowering the AIC (minimum AIC achieved with a linear model is -54 and -117 compared with a GAM of -76 and -156 for Côte d'Ivoire and Uganda respectively). The non-parametric GAM gives a better model fit with model summaries shown in (Fig 7). AIC is improved when

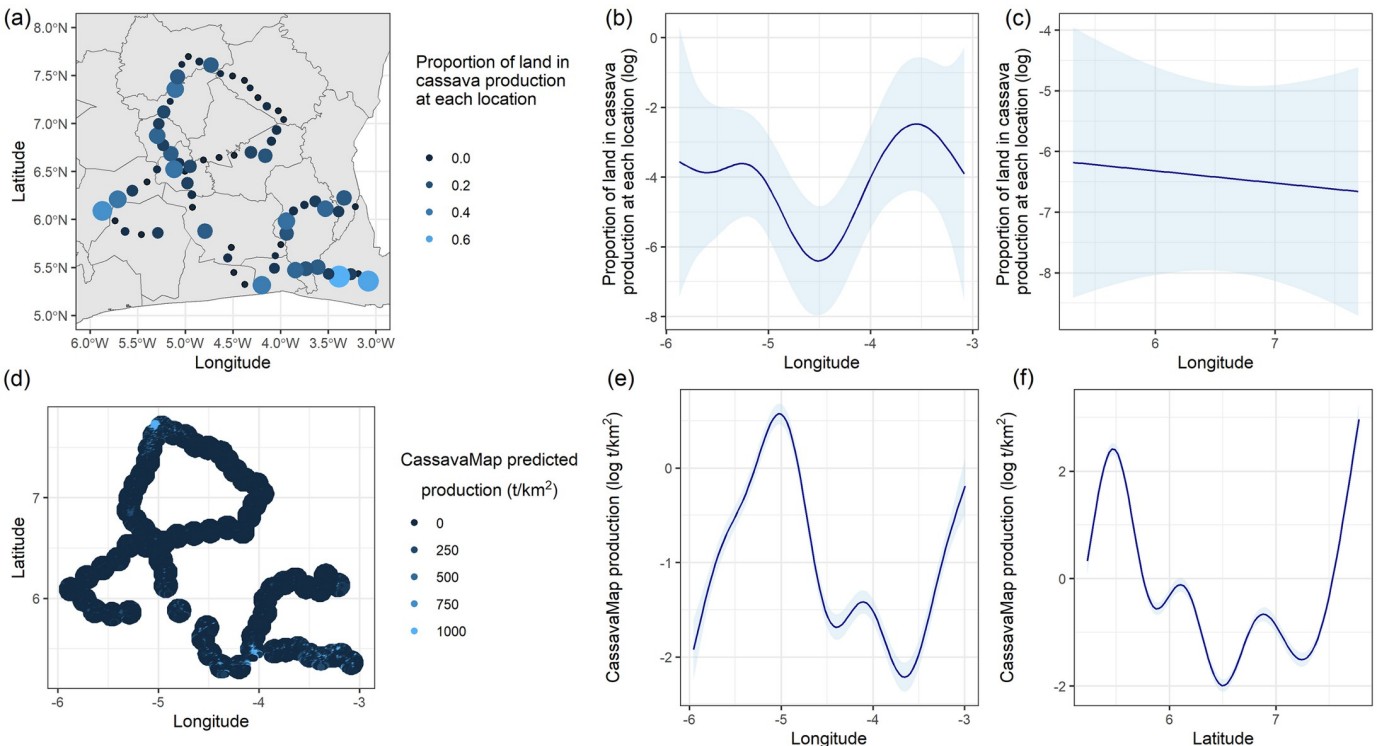

**Fig 5. Spatial trends in cassava density in Côte d'Ivoire.** a) Total cassava area (weighted) at each survey location b) and c) Predicted smooths over longitude and latitude from a fitted generalized additive model to data in a). d) Predicted Cassava production at 10km buffers from survey locations extracted from CassavaMap [14], e) and f) Estimated smooths over longitude and latitude from fitted a generalized additive model to data in c). The country shapefiles were obtained from Global Administrative Areas (GADM) [31].

cassava predictions are summarised over a buffered zone. In general, a slight difference is observed in the size of the buffer zone (2, 5 or 10 km) in Côte d'Ivoire, but in Uganda, 2 km buffer zones generally outperform larger radii. The summary type of cassava prediction has a greater influence in Côte d'Ivoire than in Uganda. Distance of buffer zone and summary type appear to have little impact on the influence of either the population or settlement summaries, although in both cases, AIC is improved (in general) when these terms are included in the model. Furthermore, in Uganda, AICs are improved when harvest area is used as predicted output from CassavaMap whilst in Côte d'Ivoire, there is no detectable difference. Interestingly, in both countries, the settlement data layer appears to have greater influence than the population data layer although both are informative (S4 and S5 Tables of S1 Appendix).

The influence of the additional data layers appears to relate to cassava cultivation. Under monoculture, more focus is given to how cassava predictions are summarised (type, distance, etc.) rather than the additional covariates whilst the opposite is seen under intercropping, with more focus on the additional data layers of population and settlement (S4 and S5 Tables of S1 Appendix). However, we cannot put too much emphasis on these results as the survey design was not stratified along these types of cultivation methods.

To visualise the non-parametric smooths fitted to the data, we fitted splines from the "best" model for each of Côte d'Ivoire and Uganda under the AIC (S2 Fig in S1 Appendix). For Côte d'Ivoire this corresponded to the following terms included as explanatory variables: the mean predicted production at a 10 km buffer, the minimum population at a 2 km buffer and the average settlement density at a 2 km buffer. The resulting splines show a relatively flat surface

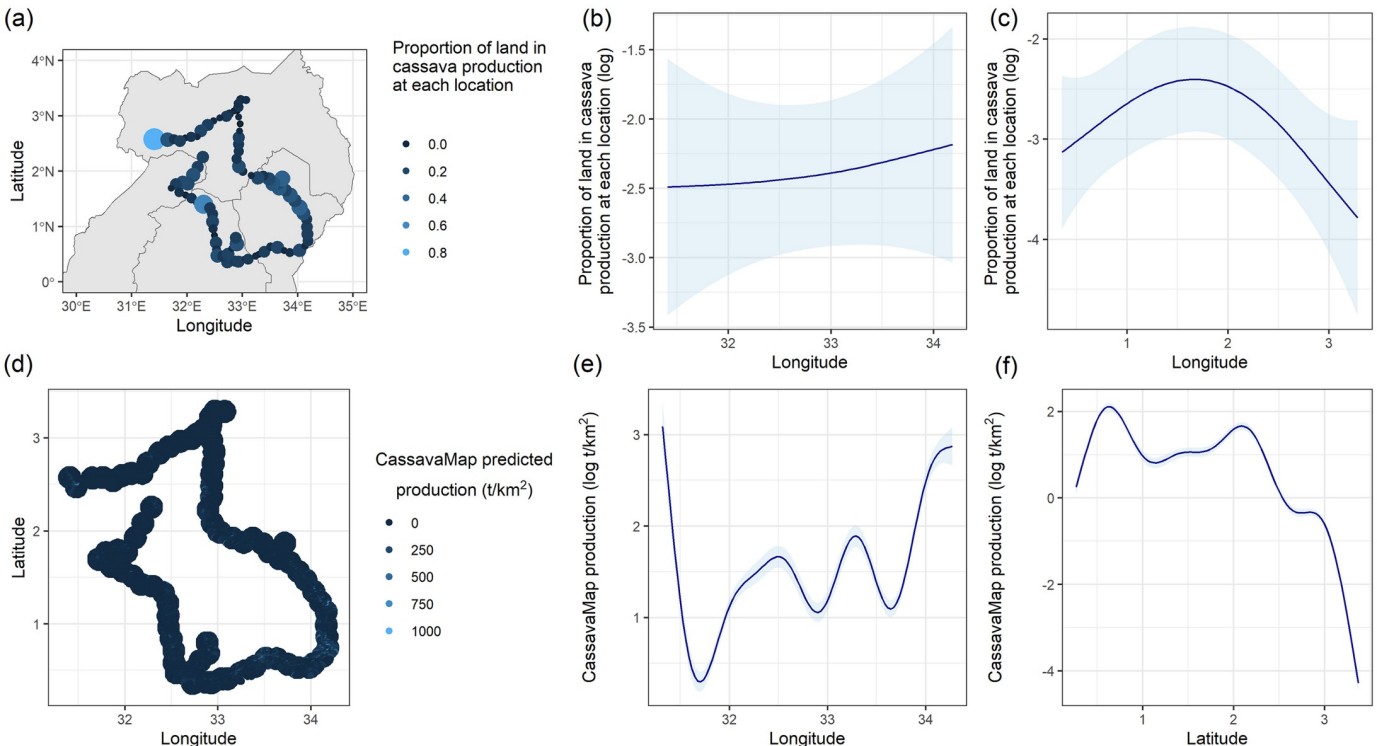

**Fig 6. Spatial trends in cassava density in Uganda.** a) Total cassava area (weighted) at each survey location b) and c) Predicted smooths over longitude and latitude from a fitted generalized additive model to data in a). d) Predicted Cassava production at 10km buffers from survey locations extracted from CassavaMap, e) and f) Estimated smooths over longitude and latitude from fitted a generalized additive model to data in c). The country shapefiles were obtained from Global Administrative Areas (GADM), [31].

fitted to the settlement layer, but a more complicated interaction between predicted cassava and population data. In particular, higher cassava production is associated with higher population values, but also medium-predicted production.

For Uganda, the "best" model corresponded to the following terms included as explanatory variables; the predicted cassava harvest area at the point location, the median population at a 5 km buffer and the average settlement density at a 2 km buffer. It is clear that a higher predicted harvest area is not necessarily associated with higher observed cassava and that this interacts on a complex surface with the population information. It can also be seen that in the average settlement density, the highest production is observed when the density is neither too sparse nor too dense.

## 4. Discussion and conclusions

In recent years substantial progress has been made in using models to identify cropland around the world, including in smallholder farming systems [32–34]. However high-precision mapping of the distribution of specific crops and their production density has lagged due to various complicating factors, including the small size of farming plots, the increased prevalence of intercropping, and crop rotation. For cassava specifically, data scarcity, highly variable agroecologies, soil conditions, and other socio-economic factors make it challenging to develop a comprehensive multivariate model to predict cassava density, despite many sources of data [2, 16, 17, 26] and models [12, 14, 15, 33] being available that may serve as indicators.

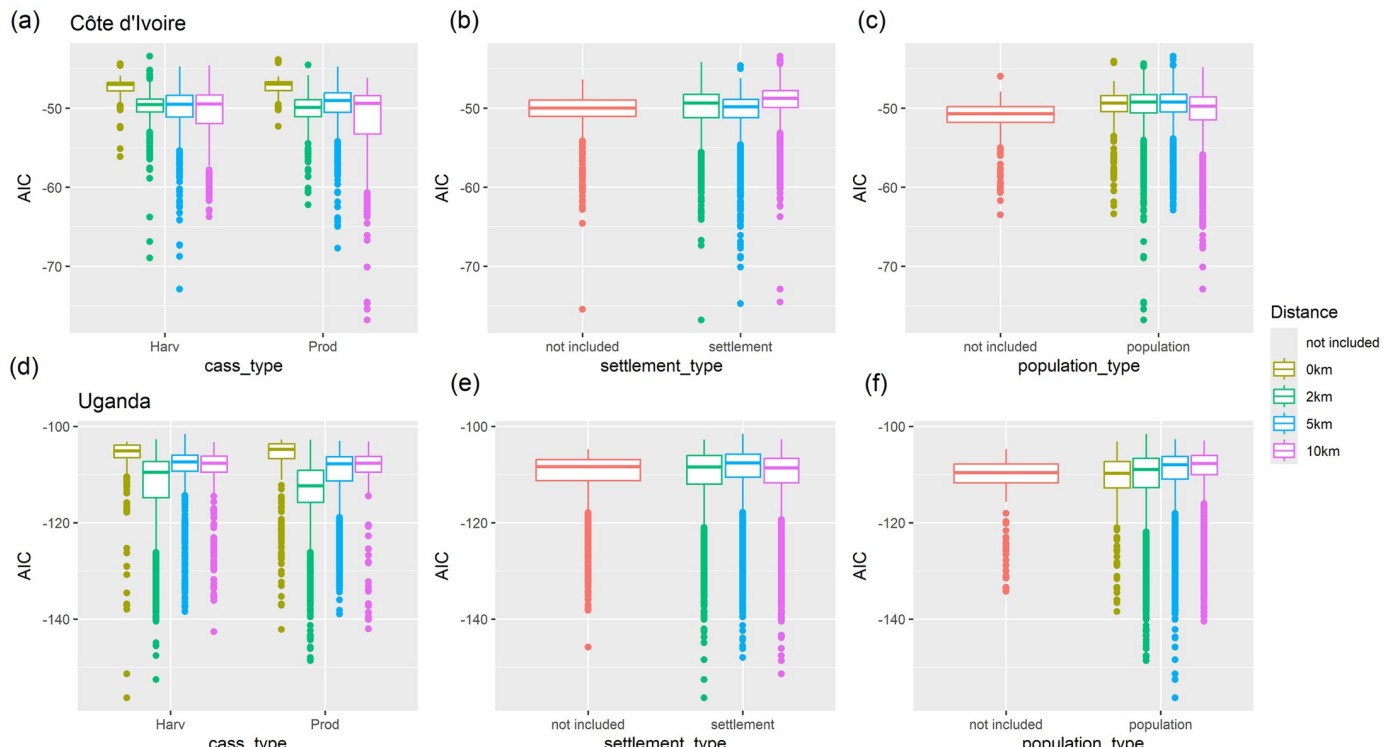

**Fig 7. AIC from all model runs using a GAM framework as outlined in Section 2.2.2 for the total cassava density response variable.** The top panel is Côte d'Ivoire and the bottom panel is Uganda.

It is expected that in rural small-holder farmer systems, the distribution of subsistence crops would be associated with the distribution of the rural populations, but this relationship may be complex. On one hand, an increase in population density increases the demand for food and calories and thus the required area under cultivation and/or the intensity of cultivation may increase. However, in high population density areas, land scarcity and consequently a gradual soil degradation can limit the area available for cultivation and therefore potential production. Additionally, in areas with high population density, alternative economic opportunities may make cassava cultivation less prevalent.

Our aim in this study was to a) quantify the characteristics of cassava cultivation across distinct cassava-growing regions, in this particular case in Uganda and Côte d'Ivoire, b) understand how well the distribution and density of non-urban populations can predict cassava density, as this had previously been considered an important cassava production predictor in sub-Saharan Africa [11, 12, 14], c) find out how the cassava density in the survey correlated with existing cassava cultivation density models and d) investigate the driving influences between the surveyed data and point predictions from CassavaMap. To test whether the relationships between the distribution and density of non-urban populations with cassava density were consistent in different regional contexts, and to discover additional links between the cassava density data collected in-situ and existing cassava cultivation models, we developed and carried out surveys in cassava-growing regions in both Uganda and Côte d'Ivoire.

Data was collected by surveying cassava fields, and collecting data on several characteristics (e.g., whether the cassava was intercropped, the planting density, etc). This allowed us to provide summaries of the characteristics of cassava cultivations in regions of Uganda and Côte d'Ivoire.

The survey demonstrated that cassava production remains an important staple crop in rural areas of Sub-Saharan Africa, with 75% and 87% of the randomly selected 200 meter square survey sites containing one or more cultivated cassava plants in Côte d'Ivoire and Uganda, respectively. However, cultivation of cassava was highly variable across sample locations both in terms of the number and size of fields but also in the type of cropping used, i.e. monoculture versus intercropping.

Baseline regression models were used to assess the association between the observed cassava production and the cassava production predicted from the CassavaMap model [14], which predicts two measures of production, the area of land under cassava cultivation and the production in each square kilometre.

Using these baseline models, we found that, in all cases, the model prediction had a non-significant relationship with the survey data, explained very little of the variation in survey data and did not establish rural population as an important driver of cassava density. However, by investigating if the presence and absence of cassava production in the surveyed locations were related to model predictions, we did find an indication of a positive relationship.

Furthermore, once we aggregated the population data, we discovered that geographical trends are present in both the survey data and the CassavaMap predictions. To associate these with geographical trends observed in the CassavaMap predictions, a buffered region about the survey locations was extracted and then generalized additive models were fitted to investigate large-scale regional changes. Despite the lack of association between the survey locations and point estimates of the model predictions, we find that at larger scales, the CassavaMap does capture large-scale variations in cassava production. It is perhaps unsurprising that model performance is improved when cassava predictions are summarised over a buffered zone as it may start to account for the spatial mismatch between a person's habitation and the location of cassava cultivation. For instance, in areas of dense population cassava fields may be located further away from the main homestead.

It is important to note that the "best" (as measured by AIC) models for observed cassava production were those that additionally included settlement and population covariate information, with the settlement data layer appearing to have greater influence than the population data layer. Furthermore, the influence of the additional data layers differs depending on the type of cassava cultivation. Under monoculture, more focus is given to how cassava predictions are summarised (type, distance, etc.) rather than the additional covariates whilst the opposite is seen under intercropping, with more focus on the additional data layers of population and settlement.

Thus, we conclude that existing models are able to capture large-scale regional trends in cassava production but fail to capture the local variation and are limited in their ability to form reliable estimates at local scale. Due to the scarcity of data, published models of cassava distribution rely on a series of assumptions to make their projections. It is evident that the cultivation of cassava in smallholder systems exhibits significant variation, likely driven by a multitude of factors ranging from climate and soil conditions to cultural preferences, and the distribution of rural population and income. Specifically, we believe that a better understanding of the drivers of cultivation practice may yield significant insight that when combined with existing models will greatly improve the accuracy of predictions of cassava production at a local scale.

Given the global importance of cassava, more comprehensive surveys linked with the application of remote sensing and machine learning, are needed to understand, upscale and model this variation across the continent and globally. Improved data collection, combined with interdisciplinary analytical approaches, will present an opportunity to better understand the

distribution of cassava spatially which will greatly benefit decision-making, cassava disease management and planning.

## Supporting information

**S1 Appendix. Validating a cassava spatial disaggregation model in sub-Saharan Africa.** (DOCX)

**S1 File. Inclusivity in global research.** (DOCX)

## Acknowledgments

We express our sincere gratitude to Richard Stutt and Lawrence Bower for their assistance in processing raw data obtained from the ArcGIS Collector for analysis and the helpful feedback on the manuscript.

## Author Contributions

**Conceptualization:** Kirsty L. Hassall, Vasthi Alonso Chávez, Anna M. Szyniszewska.

**Data curation:** Kirsty L. Hassall, Hadewij Sint, Anna M. Szyniszewska.

**Formal analysis:** Kirsty L. Hassall, Hadewij Sint, Anna M. Szyniszewska.

**Funding acquisition:** Vasthi Alonso Chávez, Anna M. Szyniszewska.

**Investigation:** Joseph Christopher Helps, Phillip Abidrabo, Geoffrey Okao-Okuja, Roland G. Eboulem, William J-L. Amoakon, Daniel H. Otron, Anna M. Szyniszewska.

**Methodology:** Kirsty L. Hassall, Vasthi Alonso Chávez, Hadewij Sint, Anna M. Szyniszewska.

**Project administration:** Vasthi Alonso Chávez.

**Supervision:** Vasthi Alonso Chávez, Anna M. Szyniszewska.

**Validation:** Kirsty L. Hassall.

**Visualization:** Kirsty L. Hassall, Hadewij Sint, Anna M. Szyniszewska.

**Writing – original draft:** Kirsty L. Hassall, Vasthi Alonso Chávez, Hadewij Sint, Anna M. Szyniszewska.

**Writing – review & editing:** Kirsty L. Hassall, Vasthi Alonso Chávez, Joseph Christopher Helps, Roland G. Eboulem, William J-L. Amoakon, Daniel H. Otron, Anna M. Szyniszewska.

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
