## [Decision Letter · Decision Letter 0]

16 Apr 2024

PONE-D-24-03273Validating a cassava spatial disaggregation model in sub-Saharan AfricaPLOS ONE

Dear Dr. Alonso Chavez,

Thank you for submitting your manuscript to PLOS ONE. After careful consideration, we feel that it has merit but does not fully meet PLOS ONE’s publication criteria as it currently stands. Therefore, we invite you to submit a revised version of the manuscript that addresses the points raised during the review process.

**Recommended Comments.**I recommend that you provide a description the programs used in the introduction and provide more information on why these programs were used.Population density is subjectively used here.  Can you provide more information as to how high, medium, and low-density population levels were defined?   More details are needed in the methods section on the locations used as suggested. For example, details on the population, area under agriculture production, and climate are required to properly define the study locations.In Table 3 please keep variable names consistent.

There seems to be a conflict with the information in Figure 4. Fig. a Scale on the X axis, “20,000” reported as 2,000.Use of a consistent scale for country comparisons.Variable names in most figures should be typed out.==============================

We look forward to receiving your revised manuscript.

Kind regards,

Angela T. Alleyne, Ph.D

Academic Editor

PLOS ONE

2. In your Methods section, please provide additional information regarding the permits you obtained for the work. Please ensure you have included the full name of the authority that approved the field site access and, if no permits were required, a brief statement explaining why."""

(2)"***Straive, at PRTC please request the following from the authors and do not ping for follow up:

Please note that PLOS ONE has specific guidelines on code sharing for submissions in which author-generated code underpins the findings in the manuscript. In these cases, all author-generated code must be made available without restrictions upon publication of the work. Please review our guidelines at https://journals.plos.org/plosone/s/materials-and-software-sharing#loc-sharing-code and ensure that your code is shared in a way that follows best practice and facilitates reproducibility and reuse.

"

(3)**Straive** Please include the following request in the decision letter, and ping me with follow up. “Please include a complete copy of PLOS’ questionnaire on inclusivity in global research in your revised manuscript. Our policy for research in this area aims to improve transparency in the reporting of research performed outside of researchers’ own country or community. The policy applies to researchers who have travelled to a different country to conduct research, research with Indigenous populations or their lands, and research on cultural artefacts. The questionnaire can also be requested at the journal’s discretion for any other submissions, even if these conditions are not met.  Please find more information on the policy and a link to download a blank copy of the questionnaire here: https://journals.plos.org/plosone/s/best-practices-in-research-reporting. Please upload a completed version of your questionnaire as Supporting Information when you resubmit your manuscript.

4. We note that [Figure(s) 1, 2, 6 a and b, 7 a and b] in your submission contain [map/satellite] images which may be copyrighted. All PLOS content is published under the Creative Commons Attribution License (CC BY 4.0), which means that the manuscript, images, and Supporting Information files will be freely available online, and any third party is permitted to access, download, copy, distribute, and use these materials in any way, even commercially, with proper attribution. For these reasons, we cannot publish previously copyrighted maps or satellite images created using proprietary data, such as Google software (Google Maps, Street View, and Earth). For more information, see our copyright guidelines: http://journals.plos.org/plosone/s/licenses-and-copyright.

1. You may seek permission from the original copyright holder of Figure(s) [1, 2, 6 a and b, 7 a and b] to publish the content specifically under the CC BY 4.0 license.  

Additional Editor Comments:

Reviewers' comments:

Reviewer's Responses to Questions

**Comments to the Author**

1. Is the manuscript technically sound, and do the data support the conclusions?

Reviewer #1: Partly

Reviewer #2: Yes

2. Has the statistical analysis been performed appropriately and rigorously? 

Reviewer #1: I Don't Know

Reviewer #2: Yes

3. Have the authors made all data underlying the findings in their manuscript fully available?

Reviewer #1: Yes

Reviewer #2: Yes

4. Is the manuscript presented in an intelligible fashion and written in standard English?

Reviewer #1: Yes

Reviewer #2: Yes

5. Review Comments to the Author

Reviewer #1: Validating a cassava spatial disaggregation model in sub-Saharan Africa

Comments and suggestions

Abstract

On line 31, include the future beneficiaries of this research

In this subheading, answer this questions why, what, how and what for

mention the importance of these studies

Introduction

Line 35, include the reference on the end of period and do this to whole document

from line 41 to 44, unbold and include the information related with cassava yield and the most cassava countries producers.

Line 49, include the reference and mention about studies/ researches related with this subject, describe the methods/ modells and explain why the authors are using this methods, I mean the advantages

line 64, why are you using references on the objective?

Mention the importance of the information of distribution of cassava cultivation, of non-urban population

mention something about satellites measurement difficult

Material and methods

Line 72, there some error, please check and correct, do the same within whole manuscript

line 73, include the reference

I suggest to divide this subheading (locations of the survey, climate and soil, socio-geografic data, farm level, remoting sensing, response variables, statistical analysis, prodution and harvest, administrative units, missing data, spacial and temporal disaggregation, data record ) in order to facilitate the explanation to the readers and avoid misunderstood.

lines 105 and 106, I suggest to include the comma instead and N, and K

line 111 to 113, merge it

line 117, rewrite

line 122, rewrite 1 kmx1Km

line 132, is it comparison or correlation

line 155 to line 163, rewrite and focus on the approach, methods that you used to collect the data

line 165, merge it

on table 3, check the last formula, is that correct?

Line 199 to 201 include the references of AIC and BIC

Results

Line 239, include the reference

line 243 to 245, the total is not 100% locations, could please check it

line 252 and 253, the total is not correct, please check it

line 275, is table 3 or 4, check it, How can we use this information? I notice thar correlation is between response variables. Clarify it,

line 309, include the Anova or in the supplementary data

line 328, include the values of AIC

Discussion

Separate this chapter with conclusions

Do not forget to compare your results with others studies

conclusion

Link or focus on the objectives

References

After consider all corrections and suggestions, cross-check the references within manuscript

Moiana, LD

Genetics and Breeding, Dr. Sc

Senior Research at IIAM/Mozambique

Scientific Coordinator at Rcol for Rice-Namacurra/Zambezia

Reviewer #2: The data presented in the manuscript supports the conclusion drawn. However, there are some areas for improvement especially in the Discussion section.

Where statistical analyses were performed, measures of variability need to be presented as well as errors for predicted and measured values in the models used.

Line 133 Which statistical package was used for the correlation analyses?

Lines 234-242  Field attributes could be placed in a table to describe the locations/countries for easier reading and comparison. Same applies for Lines 255-264. 

Line 239 Present means with standard dev. since that was mentioned as calculated in the method. Adjust throughout results section.

Line 251 Were 96 or 87 locations assessed in Uganda?

Good use of the English language.

6. PLOS authors have the option to publish the peer review history of their article (what does this mean?). If published, this will include your full peer review and any attached files.

Reviewer #1: **Yes: **Leonel Domingos Moiana

Reviewer #2: No

---

## [Author Response · Author response to Decision Letter 0]

8 Oct 2024

We thank the reviewers and editors for their time in considering our manuscript. In the document "Response to reviewers", the comments of the editor and each reviewer are addressed. We begin with the editor, reviewer 1, and finally addressing reviewers’ 2 comments.

---

## [Editor Report · Decision Letter 1]

14 Oct 2024

Validating a cassava production spatial disaggregation model in sub-Saharan Africa

PONE-D-24-03273R1

Dear Dr. Alonso Chavez,

We’re pleased to inform you that your manuscript has been judged scientifically suitable for publication and will be formally accepted for publication once it meets all outstanding technical requirements.

Kind regards,

Angela T. Alleyne, Ph.D

Academic Editor

PLOS ONE
---

## [Editor Report · Acceptance letter]

25 Oct 2024

PONE-D-24-03273R1 

PLOS ONE

Dear Dr. Alonso Chávez, 

I'm pleased to inform you that your manuscript has been deemed suitable for publication in PLOS ONE. Congratulations! Your manuscript is now being handed over to our production team.

Kind regards, 

on behalf of

Dr. Angela T. Alleyne 

Academic Editor

PLOS ONE